# On the Cleanliness of Different Oral Implant Systems: A Pilot Study

**DOI:** 10.3390/jcm8091280

**Published:** 2019-08-22

**Authors:** Dirk U. Duddeck, Tomas Albrektsson, Ann Wennerberg, Christel Larsson, Florian Beuer

**Affiliations:** 1Department of Prosthodontics, Geriatric Dentistry and Craniomandibular Disorders, University Charité Berlin, 14197 Berlin, Germany; 2CleanImplant Foundation, Research Department, 10117 Berlin, Germany; 3Department of Biomaterials, Institute for Clinical Sciences, Sahlgrenska Academy, University of Gothenburg, 40530 Gothenburg, Sweden; 4Department of Prosthodontics, Institute of Odontology, The Sahlgrenska Academy, University of Gothenburg, 40530 Gothenburg, Sweden; 5Department of Prosthodontics, Faculty of Odontology, Malmö University, 20506 Malmö, Sweden

**Keywords:** dental implants, surface properties, titanium, materials testing, implant contamination, implant surface, scanning electron microscopy, energy-dispersive X-ray spectrometry

## Abstract

(1) Background: This paper aimed to compare the cleanliness of clinically well-documented implant systems with implants providing very similar designs. The hypothesis was that three well-established implant systems from Dentsply Implants, Straumann, and Nobel Biocare were not only produced with a higher level of surface cleanliness but also provided significantly more comprehensive published clinical documentation than their correspondent look-alike implants from Cumdente, Bioconcept, and Biodenta, which show similar geometry and surface structure. (2) Methods: Implants were analyzed using SEM imaging and energy-dispersive X-ray spectroscopy to determine the elemental composition of potential impurities. A search for clinical trials was carried out in the PubMed database and by reaching out to the corresponding manufacturer. (3) Results: In comparison to their corresponding look-alikes, all implants of the original manufacturers showed—within the scope of this analysis—a surface free of foreign materials and reliable clinical documentation, while the SEM analysis revealed significant impurities on all look-alike implants such as organic residues and unintended metal particles of iron or aluminum. Other than case reports, the look-alike implant manufacturers provided no reports of clinical documentation. (4) Conclusions: In contrast to the original implants of market-leading manufacturers, the analyzed look-alike implants showed significant impurities, underlining the need for periodic reviews of sterile packaged medical devices and their clinical documentation.

## 1. Introduction

The advent of osseointegration has led to a clinical breakthrough in oral implants. Minimally rough, turned implants were the first osseointegrated oral implants used, with the first patient treated in 1965 [1]. These turned screws remain the most clinically documented implants of all with 75% of all studies in long-term reports [2]. Over time, other clinically documented oral implant systems have increasingly begun to be used. Those systems may have preferred slightly different surfaces; moderately rough implants have been the treatment of choice since the turn of the millennium, since they have demonstrated improved clinical results [3]. The surface of moderately rough implant systems may be manufactured in different ways, by subtractive methods such as the combination of blasting and acid etching or anodization and by additive techniques, such as hydroxyapatite coating. Products from what are regarded as the three largest oral implant companies in the world including Osseo Speed implants from Dentsply–Sirona, SLA-implants from Straumann and TiUnite implants from Nobel Biocare, have been clinically documented in numerous papers spanning a period of over 5 to over 10 years of follow up with very high levels of survival and success [4,5,6].

Since some osseointegrated oral implant systems have been duly documented with a very good clinical outcome, numerous new implant manufacturers have tried to mimic the surfaces and geometries presented by the leading companies. These are so called copy-cat or at least “look-alike” implant systems which usually lack clinical documentation of their own but claim to be as good as the original implants they are trying to mimic. However, in clinical reality these implants lack the scientific evidence of similar performance.

One surface characteristic of sterile packaged oral implants is their cleanliness. Oral implants may display different surface of an inorganic or organic nature. These impurities may derive from the manufacturing handling and packaging processes and may remain on the commercially available implant. We are presently lacking in knowledge of the precise clinical risks of implant impurities. However, contaminations are technically avoidable and, generally speaking, the authors assume all of us would prefer clean implants to avoid potential problems from surface impurities.

The aim of the present paper was to compare the cleanliness of proper clinically documented implant systems with implants that are very similar in design and surface; OsseoSpeed from Dentsply Implants was compared to a German look-alike implant called Cumdente; Standard Plus Implant SLA from Straumann was compared to a Chinese look-alike implant called Bioconcept (claiming to be 100% compatible with Straumann); and a TiUnite-surfaced implant NobelActive from Nobel Biocare was compared to a Swiss/Taiwanese implant system called Biodenta. Our hypothesis was that the three major and well-established implant systems have significantly comprehensive clinical documentation and have their implants produced in a significantly cleaner manner than would the respective look-alike systems.

Every single dental implant has to be clean, as this is a medical device that could harm patients—even if we found one single implant with impurities, this implant was intentionally sold for the therapy of one real patient. This paper was not intended to show a statistically relevant number of average contaminations for specific implant types. All implants in this study were randomly purchased and labeled for clinical use. Each sample of these medical devices was produced using a certain regime of quality management. If the quality management of a manufacturer cannot ensure a certain level of cleanliness, a single implant with significant impurities, which are technically avoidable, is proof of a lack of quality.

## 2. Experimental Section

Implant types used for this analysis were the following: Astra Tech–Dentsply Implants (OsseoSpeed EV, Mölndal, Sweden), Straumann (Standard Plus SLA Implant, Zürich, Switzerland), Nobel Biocare (NobelActive Internal RP, Zürich, Switzerland), Cumdente (AS Implant, Tübingen, Germany), Bioconcept (Tissue Level Implant, Jiangsu, China), Biodenta (Dental Implant, Bernek, Switzerland). The three implants from market-leading manufacturers and the correspondent three look-alike implants were purchased in the period between March 2018 and May 2019, either by ghost-shopping, where the ordering practice was reimbursed from the research fund or by direct order. In all six cases, the manufacturers or the respective distributors were not informed about the purpose of the implant order. None of the samples were provided free of charge. Prices varied from 191 euro to 322 euro for the samples of market-leading brands and from 78 euro to 276 euro for the look-alike products.

All of the six samples collected were carefully unpacked, mounted on the sample holder on carbon tabs without touching the implant surface, and analyzed with a scanning electron microscope (SEM) in a particle-free clean room environment (according Class 100 US Federal Standard 209E, Class 5 DIN EN ISO 14644-1) to avoid artifacts from the ambient air (Figure 1).

The scientific workstation used was a Phenom proX Scanning Electron Microscope (Eindhoven, Netherlands), equipped with a high-sensitivity backscattered electron (BSE) detector. The detector for the energy-dispersive X-ray spectroscopy (EDS) and elemental analysis was a thermoelectrically cooled silicon drift detector (SDD) type, with an active detector area of 25 mm^2^.

The high-sensitivity BSE detector allows a magnification of up to 100,000× with a resolution down to 15 nm. This study used material-contrast images from 500× to a magnification of 5000×. Material-contrast imaging gave additional information about the chemical nature and allocation of different remnants or contaminations on the sample material.

In order to achieve a complete overview of the horizontally mounted implant sample and comprehensive surface quality information in high resolution, implants were scanned at a magnification of 500× in the “Image-Mapping” mode prior to the detailed analysis of potential impurities. This technique produces up to 600 single high-resolution SEM images of the implant surface that were digitally composed into one large image, with an extremely high resolution. The composed SEM image, showing the full size of the implant from shoulder to apex, made it possible to count particles in the visible field (viewing angle of approximately 120°) and to identify areas of interest for a subsequent EDS spot analysis. After the mapping process, SEM images of impurities and other regions of interest were produced with 500×, 1000× 2500×, and 5000× magnification. In the next step, the elemental composition of particles was determined and, where possible, the differential spectra of particles were achieved to subtract signals from the core material and such focus on signals from the superficial contamination (Figure 2).

All of the analyses, as well as the complete setup, as described above were performed at the Medical Materials Research Institute, Berlin, Germany, which is an officially accredited (Deutsche Akkreditierungsstelle–DAkkS) and externally audited testing laboratory according to the international standards DIN EN ISO 9001:2015, ISO 22309:2015 and DIN EN ISO/IEC 17025. These standards were chosen as a precondition in order to assure testing procedures at the highest level of accuracy.

In addition to the SEM/EDS analysis, all of the implants in this study were provisionally evaluated from a surface topographical point of view by interferometry. All implants seemed to be in the moderately rough surface range, i.e., with Sa (Sa = arithmetical mean height of the surface) values of between 1 and 2 micrometers.

### 2.1. Clinical Documentation of Analyzed “Look-Alike” Implant Systems

A search for available clinical trial regarding the dental implant systems was carried out. Initially the website of each dental implant manufacturer was searched (www.biodenta.com, www.bioconcept.cn, www.cumdente.com). In addition, the manufacturers were contacted via their respective contact e-mail address on their websites, requesting any scientific documentation regarding clinical performance such as published papers or summaries of ongoing projects. If no response was received within one week, a reminder was sent.

Furthermore, a search for clinical trials was performed in the PubMed database (PubMed.gov, US National Library of Medicine, National Institutes of Health). The search terms “dental implants” (MeSH) and “dental implants” (free text) were used in combination with the product name “Biodenta”, “Bioconcept”, and “Cumdente”. No limits were set. ((“dental implants” [MeSH Terms] OR (“dental” [All Fields] AND “implants” [All Fields]) OR “dental implants” [All Fields]) AND biodenta [All Fields]), ((“dental implants” [MeSH Terms] OR (“dental” [All Fields] AND “implants” [All Fields]) OR “dental implants” [All Fields]) AND bioconcept [All Fields], ((“dental implants” [MeSH Terms] OR (“dental” [All Fields] AND “implants” [All Fields]) OR “dental implants” [All Fields]) AND cumdente [All Fields])).

### 2.2. Clinical Documentation of OsseoSpeed, SLA, and TiUnite Implant Systems

With respect to the implant systems OsseoSpeed from Dentsply-Sirona, SLA from Straumann, and TiUnite from Nobel Biocare, these belong to the most clinically documented oral implant systems in the world [2]. To remain brief, we decided to only quote five papers for each system as the total number of clinical reports on these devices amounts to several hundred scientific papers.

## 3. Results

### 3.1. SEM Imaging and Elemental Analysis

Implants were analyzed in three groups. In the first group, the implant from Astra Tech–Dentsply Implants (OsseoSpeed) and the implant from Cumdente (AS Implant) were compared. The full-size SEM image of the Astra Tech implant—digitally composed of 455 single SEM images (tiles)—showed a homogenous surface with no foreign material (Figure 3).

Higher magnification could identify the TiO_2_ particles from the blasting process seen as sharp-edged particles of 5–10 mm embedded in the titanium surface. Elemental analysis of these particles only displayed signals of titanium and oxygen (Figure 4).

The Cumdente implant showed several anomalies in the correspondent full-size image, composed of 422 tiles (Figure 5).

The SEM images with higher magnification revealed systematic contamination with multiple (>100) organic particles (5–60 µm) on exposed parts of the implant, as seen on the micro-threads next to the implant shoulder (Figure 6) and near the implant’s apex (Figure 7).

In order to receive information about the particles’ elemental composition, EDS measurement was performed with a spot focused on the foreign material, where background material was always detected as well, and another spot focused in direct proximity, where only the implant’s core material was detected (spot #1 and #2 in Figure 6b and Figure 7b, respectively). Using a software application, it was possible to subtract the signals of the core material so that the differential measurement revealed more precise information about the elemental composition of the foreign material. Figure 8 shows the differential EDS measurement of the particle in Figure 7, with a clear signal of carbon as the major element in this impurity.

One larger area at the implant shoulder showed numerous particles (5–40 µm) (Figure 9) with significant signals, not only of carbon but also of fluorine, as seen in the differential EDS analysis (Figure 10). The texture and elemental composition of the foreign material suggest that these particles are most likely remnants of polytetrafluoroethylene (PTFE), used at different implant production stages.

The second group compared the Straumann Standard Plus SLA Tissue Level Implant with an implant of the same geometry from Bioconcept, both made of commercially pure grade 4 titanium, which is a composition of 99% titanium, 0.50% iron (maximum), oxygen 0.40% (maximum), carbon 0.085 (maximum), hydrogen 0.015% (maximum), and nitrogen 0.05% (maximum), according to the ASTM F67 specification. The SEM imaging of the Straumann implant could not detect any organic or inorganic contaminants (Figure 11 and Figure 12).

While the Straumann implant can be rated as clean at the micron level, the analysis of the Bioconcept implant revealed several impurities, although the full-size image showed no systematic larger contamination (Figure 13). With higher magnification, approximately 20 small organic particles (10–20 µm) were found on the implant’s surface, one particle showing additional signals of sulfur (Figure 14).

Surprisingly, one metal particle of 8 µm was found on the Bioconcept implant, seemingly entirely composed of iron, as the qualitative and quantitative elemental analysis and differential measurement of this particle revealed (Figure 15 and Figure 16).

The third group compared the NobelActive implant from Nobel Biocare with the Dental Implant from Biodenta. Both implants have an anodized surface with a characteristic titanium oxide layer. The area analysis of the NobelActive implant showed calcium and phosphorous signals in addition to titanium and oxygen. Within the scope of this analysis, neither inorganic nor organic particles were detected on the Nobel Biocare implant (Figure 17 and Figure 18).

While the full-size SEM image of the Biodenta implant exposed no significant organic contamination (Figure 19), the area analysis of the core material showed, in addition to calcium, phosphorous, titanium, and oxygen, high levels of magnesium in the EDS (Figure 20) that were not seen on the Nobel Biocare implant.

Significant traces of aluminum were detected in three areas of exposed threads and can be seen as bright particles (5–10 µm) in Figure 21. The graph in Figure 22 shows the correspondent EDS differential measurement of metal particles in Figure 20a.

These aluminum-containing particles must be rated as inorganic impurities as they are not comparable with sharp-edged aluminum oxide particles, used for blasting procedures of other implants. Apart from this, it is noticeable that the approximately 3 µm thick oxide layer was damaged at several exposed locations of the Biodenta implant (Figure 23).

### 3.2. Documentation of Clinical Results

None of the copy-cat manufacturers’ websites contained any information regarding published clinical trials of the implant systems. No manufacturers responded to the email requesting scientific documentation regarding clinical performance. No manufacturers presented such documentation. The PubMed database search identified no papers regarding the Bioconcept or Cumdente system, but two papers for the Biodenta system. The two Biodenta papers were, however, in vitro trials that evaluated the company’s CAD/CAM (computer-aided design/computer-aided manufacturing) system, not the clinical outcome of the implants [7,8]. Broadening the search by deleting “dental implants” and only using the respective company name did not identify any publications for the Cumdente system but one additional paper for the Biodenta system [9]. However, this paper did not report on the clinical outcome of implants. No clinical documentation of implant outcome was noticed for the Bioconcept system.

The OsseoSpeed implant has a very solid clinical documentation verifying excellent clinical results for up to 7 years in clinical function [10,11,12,13,14]. Similar excellent clinical results apply to the SLA implants from Straumann [4,15,16,17,18] and the TiUnite implants from Nobel Biocare [5,19,20,21,22], where these two implant systems have been documented for more than 10 years in numerous studies.

## 4. Discussion

In general, the implants OsseoSpeed (Astra Tech–Dentsply Implants), Standard Plus SLA Implant (Straumann), and NobelActive (Nobel Biocare) showed—within the scope of this analysis—a surface free of foreign materials in the SEM. The Cumdente AS Implant with a similar geometry compared to the Astra Tech OsseoSpeed implant demonstrated a surface with substantial organic contaminants and most likely remnants of Teflon. The Straumann look-alike implant from Bioconcept exposed numerous organic particles and two small particles containing significant amounts of iron. The Nobel Biocare look-alike implant from Biodenta showed high levels of magnesium and small particles with aluminum on the surface. These particles, with a diameter of 5 to 10 µm, and organic contaminants with a similar size, are small enough for phagocytosis by macrophages that would be theoretically possible for particles without a chemical bonding to the implant surface.

It may, therefore, be said that the original implants were cleaner than the correspondent look-alike devices. The price range for the look-alike implants indicated that the definition of a copy-cat product or look-alike implant is not necessarily based on a low price.

To the knowledge of the present authors, no defined thresholds have been published in peer reviewed journals until now, with respect to what may represent “acceptable” levels of impurities and what must be regarded as “unacceptable impurities”. However, the non-profit CleanImplant Foundation (www.cleanimplant.org) has presented a consensus statement on surface impurities signed by Luigi Canullo (Rome, Italy), Jaafar Mouhyi (Marrakesh, Morroco), Michael Norton (London, UK), and four of the authors of this paper Tomas Albrektsson, Florian Beuer, Dirk Duddeck, and Ann Wennerberg [23]. In this consensus statement, surface anomalies and remnants of blasting materials were not considered clinically relevant, in contrast to the metal particles of tungsten, nickel, iron, chromium, copper, tin or antimony found on the surfaces of some implants. Single organic particles smaller than 50 µm in diameter were considered less vicious than numerous particles, with a maximum of 30 particles along the circumference of the implant. Major plaque-like organic contaminants exceeding the size of 50 µm and PTFE particles, presumably originating from Teflon baskets used during implant production, were considered unacceptable.

Other foreign bodies routinely seen adjacent to implants such as titanium particles [24,25,26] or the accidental presence of cement in the bone-to-implant interface that, according to some investigators, may be found in 59% of cemented implants [27], may combine to cause peri-implantitis [28].

Impurities on sterile packaged implants—caused by metal particles and contaminations with organic substances such as thermoplastic materials, synthetic polymers, or polysiloxanes—are technically avoidable, as this paper demonstrated. The academic discussion as to what extent implant pollution is acceptable normally ends quickly when dental professionals know about such contamination of an implant system and the next patient for an implant therapy is their partner or child. We should avoid using sterile packaged implants with verifiable impurities and therewith follow the well-established “precautionary principle” as an evolution of the ancient medical principle of “primum non nocere”.

From a clinical point of view, it is obvious that Astra Tech–Dentsply Implants, Straumann, and Nobel Biocare have solidly documented clinical results reported in hundreds of scientific papers. On the other hand, a long history of clinical documentation representing high-quality efforts in the past is no guarantee for a high level of production quality and a clean surface at present, accentuating the need for periodic reviews by independent institutions. With respect to the look-alike systems, none of them had any properly documented clinical reports available. The absence of clinical documentation of look-alike systems may indicate either that they see themselves as identical to the implants they have copied or that they see implants as commodity products where all implant systems will work well clinically. The possibility first mentioned was critically analyzed in this paper comparing the cleanliness of different implant systems. The second possibility that implants are commodity products may be criticized against the knowledge of numerous implant systems that have been withdrawn from the market due to the unforeseen clinical problems over longer times of follow-up [29].

In essence, our hypothesis was verified in that the major implant systems displayed cleaner surfaces than those of their respective look-alike implants. However, whether the relative lack of cleanliness of look-alike implants indicates that they have an impaired clinical function in comparison to the major systems remains uncertain. Having said this, oral implants are placed in human beings and, therefore, it seems strongly advisable to present clinical results in peer-reviewed journals for every oral implant system to be used clinically. In this regard, the look-alike implants are clearly inferior to the major documented oral implant systems and none of the look-alike systems investigated had any clinical documentation of their own, which must be regarded as a major shortcoming. Since differences obviously exist between the major systems and the respective look-alike implants, clinicians using the latter devices must inform their patients of this fact and that the implants placed are totally un-documented with respect to clinical outcome.

## 5. Conclusions

In contrast to the original implants of market-leading manufacturers, the analyzed look-alike implants showed significantly more impurities, underlining the need for periodic reviews of the production quality by independent institutions. Multiple organic particles and remnants of PTFE (Cumdente), organic particles containing sulfur, particles containing iron (Bioconcept) or impurities with aluminum (Biodenta)—all particles small enough for possible phagocytosis—expose patients to unknown risks. In addition to the results of SEM/EDS analysis, the lack of clinical documentation of the analyzed look-alike implants raises concerns.

## Figures and Tables

**Figure 1 jcm-08-01280-f001:**
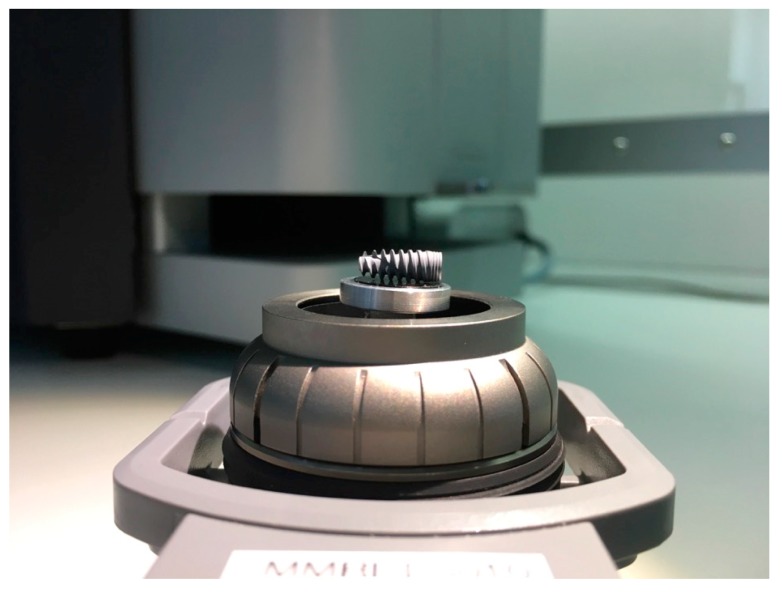
Implant sample with a length of 10 mm mounted on the SEM sample holder.

**Figure 2 jcm-08-01280-f002:**
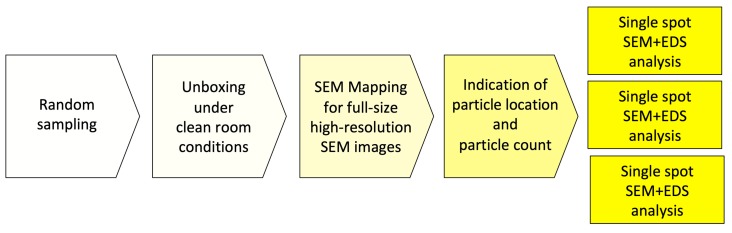
Workflow of the SEM/EDS analysis.

**Figure 3 jcm-08-01280-f003:**
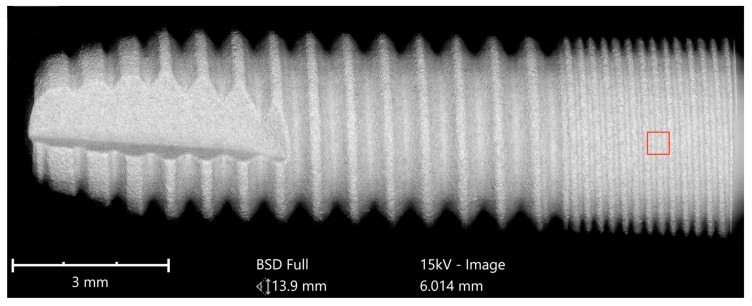
SEM mapping of OsseoSpeed implant (Astra Tech–Dentsply Implants). Magnification of the red marked area is shown in Figure 4.

**Figure 4 jcm-08-01280-f004:**
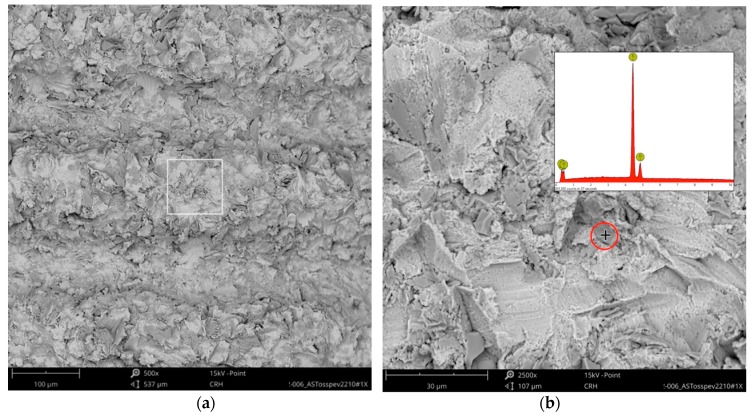
OsseoSpeed surface: (**a**) the red marked area in Figure 3 magnified 500×; (**b**) higher magnification (2500×) of the white marked area in the left image with EDS spot analysis of an embedded TiO_2_-particle (spot is marked with “+” in the red circle) showing only signals of the blasting material.

**Figure 5 jcm-08-01280-f005:**
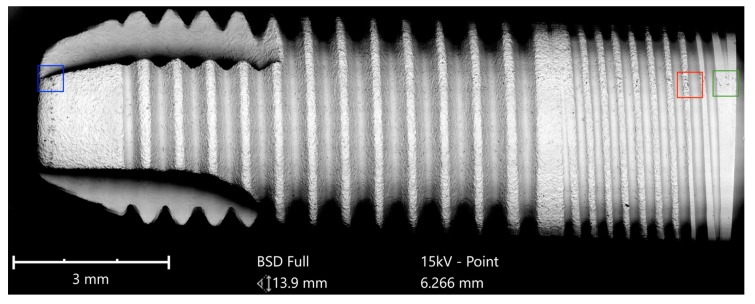
SEM mapping of the AS Implant (Cumdente); Red marked area—see magnification in Figure 6, blue marked area—see magnification in Figure 7, green marked area—see magnification in Figure 8.

**Figure 6 jcm-08-01280-f006:**
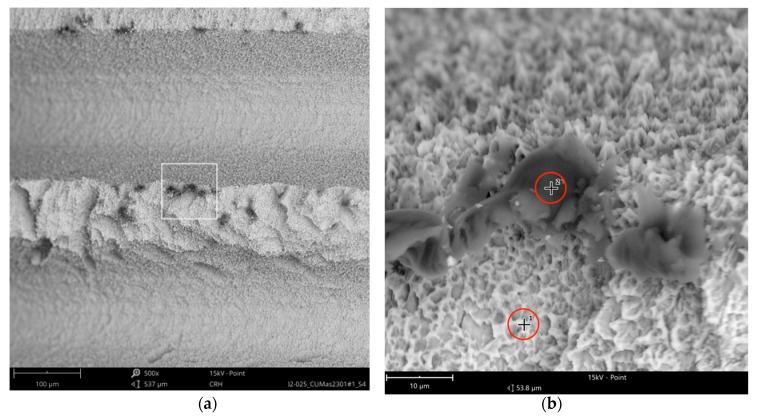
AS Implant (Cumdente) with organic particles (10–50 µm) at the implant shoulder: (**a**) systematic contamination of exposed threads, red marked area of Figure 5 in 500×; (**b**) magnification (5000×) of white marked area in the left image. The EDS differential measurement of the marked spots was identical to the particles in Figure 7.

**Figure 7 jcm-08-01280-f007:**
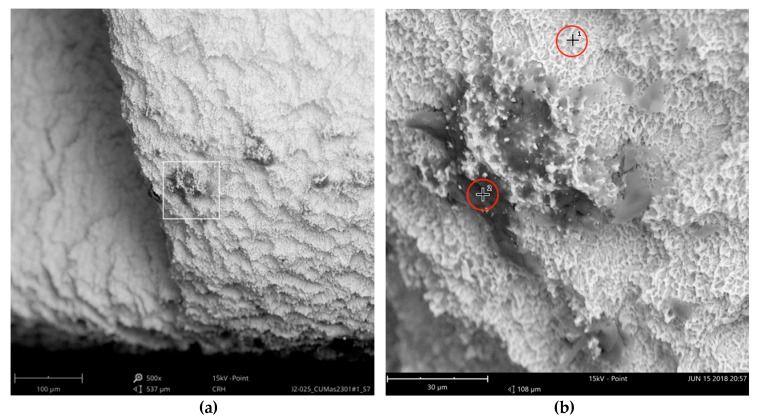
AS Implant (Cumdente) with organic particles (5–70 µm) at the implant’s apex: (**a**) blue marked area of Figure 5 in 500×; (**b**) magnification (5000×) of white marked area in Figure 7a; EDS differential measurement of marked spots is shown in Figure 8.

**Figure 8 jcm-08-01280-f008:**
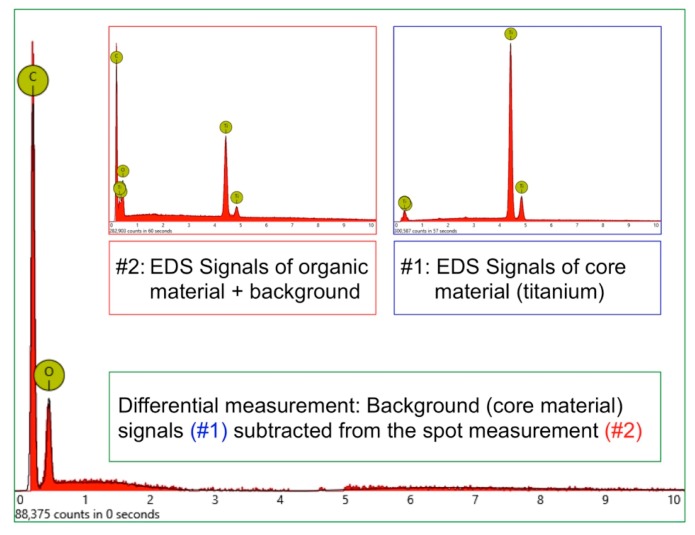
EDS differential measurement of the organic particle in Figure 7. Graph #1 shows titanium as the core material of this implant, with characteristic peaks at 0.452 KeV (La), 4.508 KeV (Lb), and 4.932 KeV (Kb). Graph #2 shows the spectrum of the particle with additional signals of the titanium background. The subtraction of X-ray quanta from the background material reveals the elemental composition of the impurity, which is carbon (characteristic Ka peak at 0.277 KeV) and oxygen (Ka peak at 0.523 KeV).

**Figure 9 jcm-08-01280-f009:**
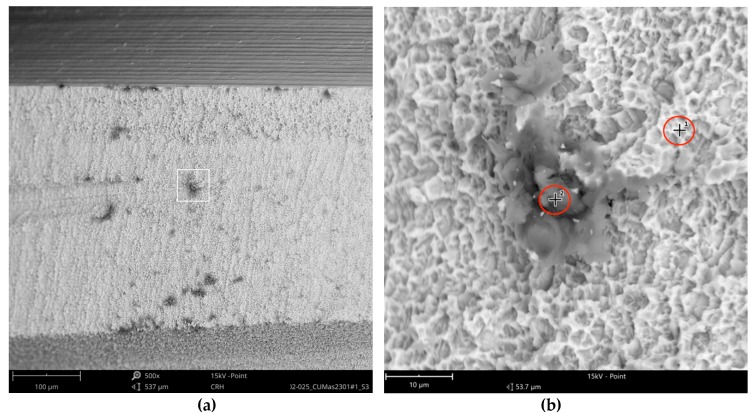
Possible remnants of polytetrafluoroethylene (PTFE), AS Implant (Cumdente); (**a**) green marked area of Figure 5 in 500× shows a larger area of impurities; (**b**) magnification (5000×) of white marked area in the left image; EDS differential measurement of the marked spots is shown in Figure 10.

**Figure 10 jcm-08-01280-f010:**
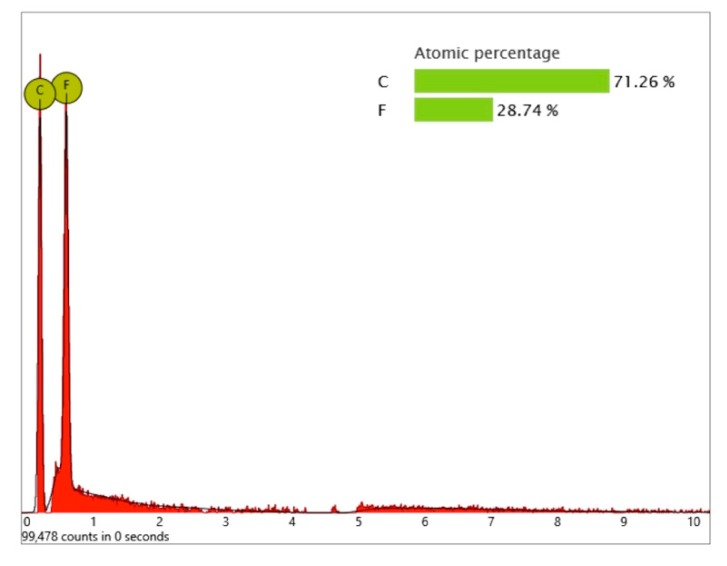
EDS differential measurement of the particles in Figure 9 with significant X-ray quanta of carbon and fluorine. Note that the quantitative information provided by the EDS analysis does not always reflect the precise stoichiometric relationship of the particle’s chemical elements.

**Figure 11 jcm-08-01280-f011:**
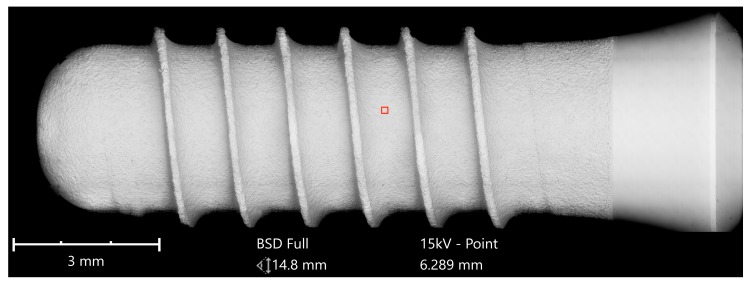
SEM mapping, Standard Plus SLA implant (Straumann); Magnification of red marked area is shown in Figure 12.

**Figure 12 jcm-08-01280-f012:**
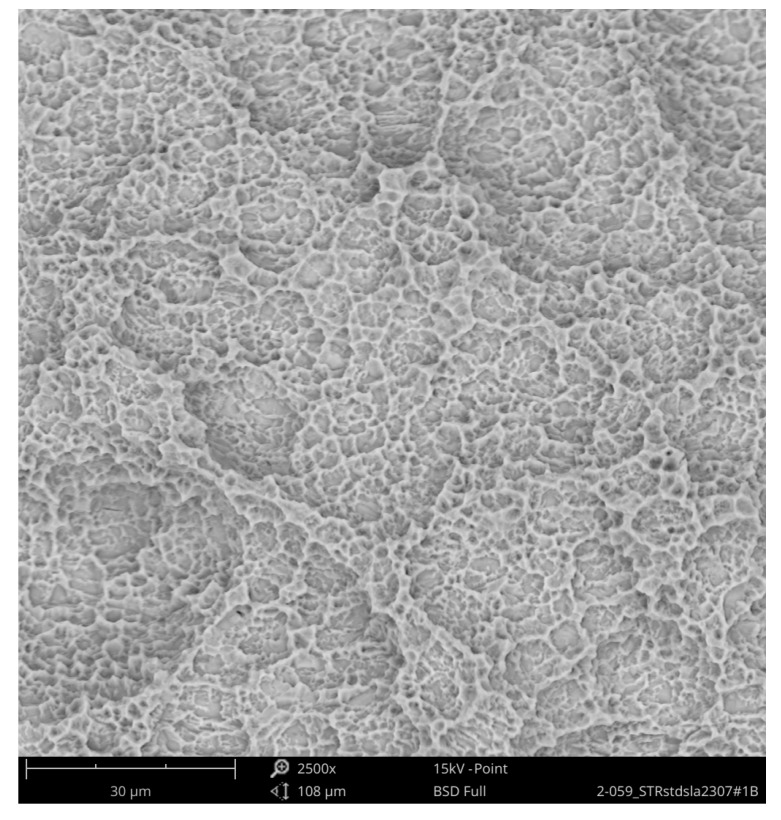
Surface of Standard Plus SLA implant (SEM 2500×); red marked area in Figure 11 with the typical texture of a sandblasted and acid-etched titanium surface, free of foreign materials.

**Figure 13 jcm-08-01280-f013:**
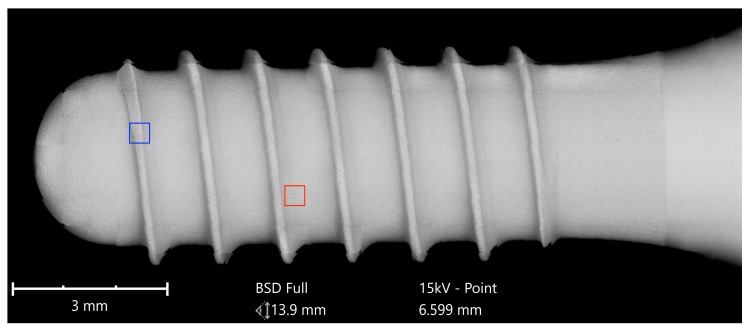
SEM mapping, Tissue Level Implant (Bioconcept); red marked area—see magnification in Figure 14, blue marked area—see magnification in Figure 15.

**Figure 14 jcm-08-01280-f014:**
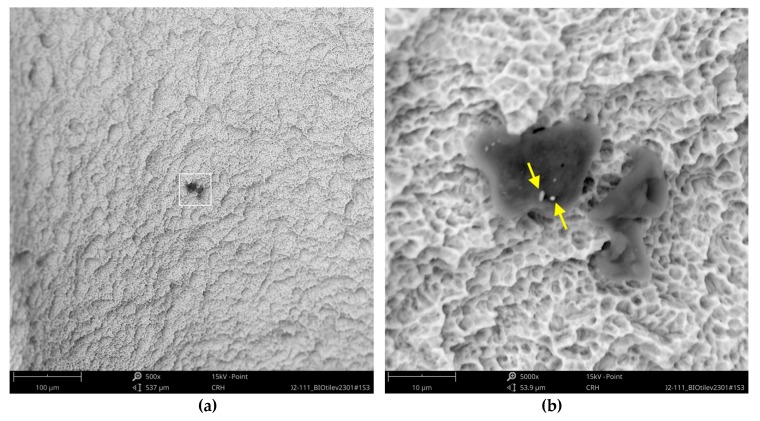
Bioconcept implant with organic particles (10–15 µm); (**a**) red marked area in Figure 13, SEM 500×; (**b**) magnification (5000×) of white marked area in the left image (**a**); arrows indicate two embedded metal particles (0.5–1 µm) showing significant iron signals in the subsequent EDS differential measurement.

**Figure 15 jcm-08-01280-f015:**
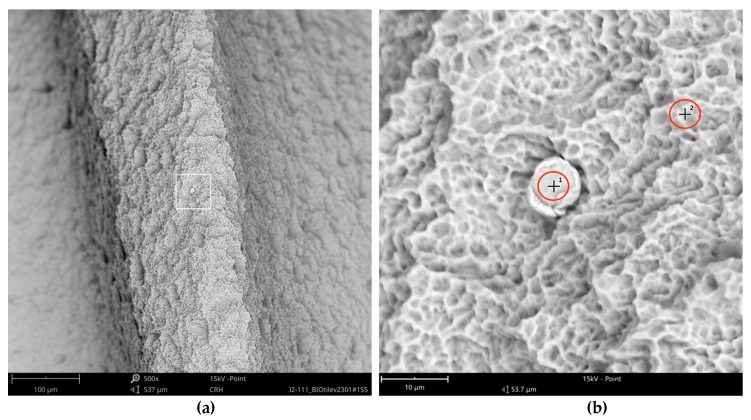
Metal particle on the surface of the Bioconcept implant; (**a**) blue marked area in Figure 13, SEM 500×; (**b**) magnification (5000×) of white marked area in the left image (**a**) showing bright metal impurity (8 µm) and 2 points of EDS spot measurement (#1 = metal particle, #2 = core material).

**Figure 16 jcm-08-01280-f016:**
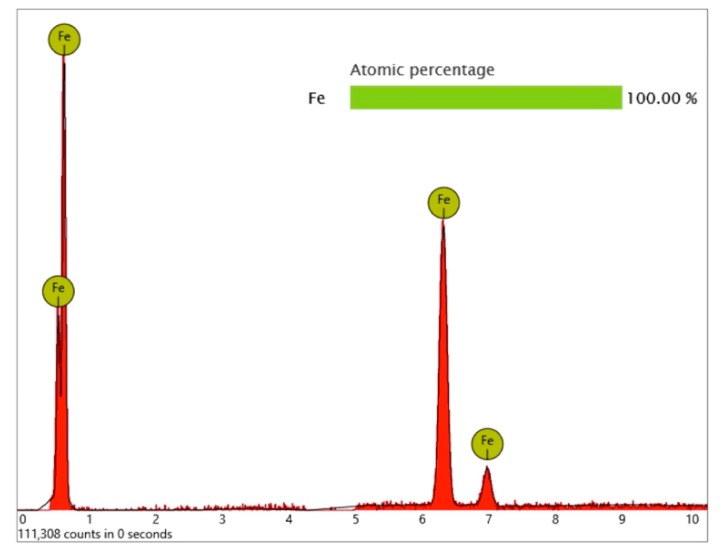
EDS differential measurement of the metal particle in Figure 15, revealing iron as the major element of the impurity.

**Figure 17 jcm-08-01280-f017:**
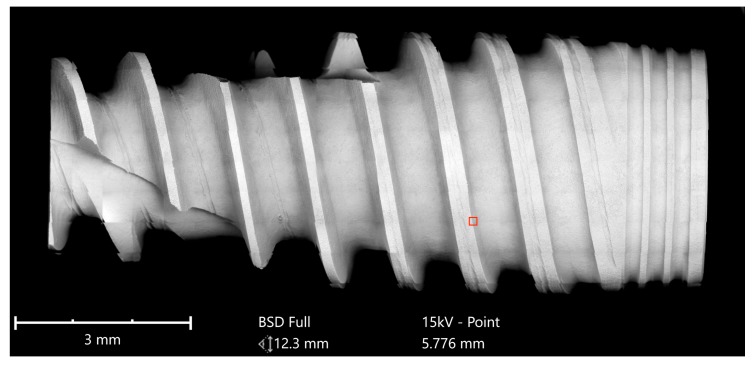
SEM mapping: NobelActive Internal RP implant (Nobel Biocare); red marked area—see magnification in Figure 18.

**Figure 18 jcm-08-01280-f018:**
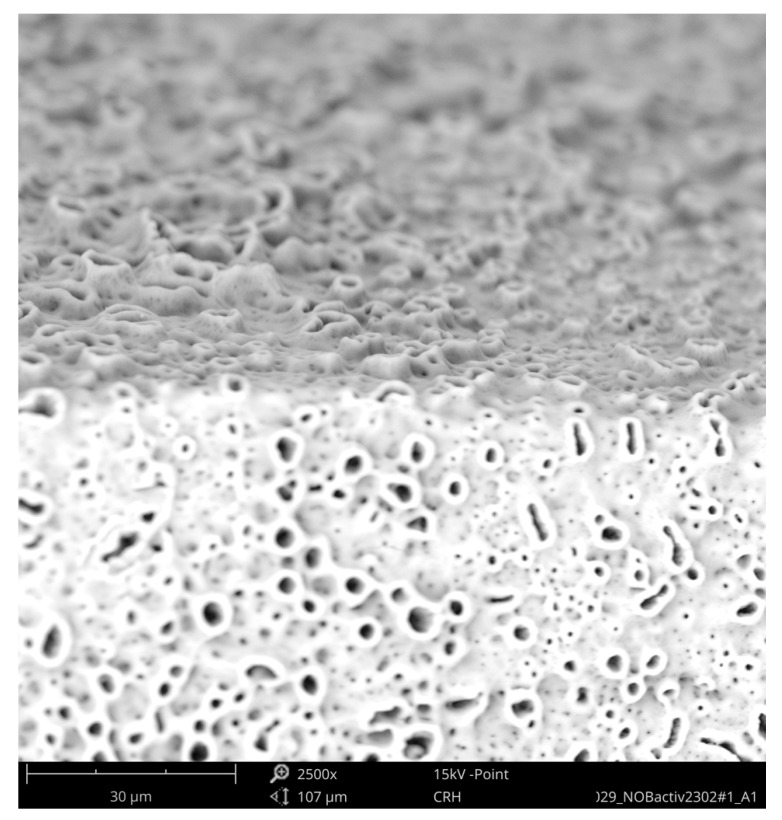
Surface of the NobelActive Internal RP implant (2500×); magnification of red marked area in Figure 17 with the typical texture of an anodized titanium surface, free of foreign materials.

**Figure 19 jcm-08-01280-f019:**
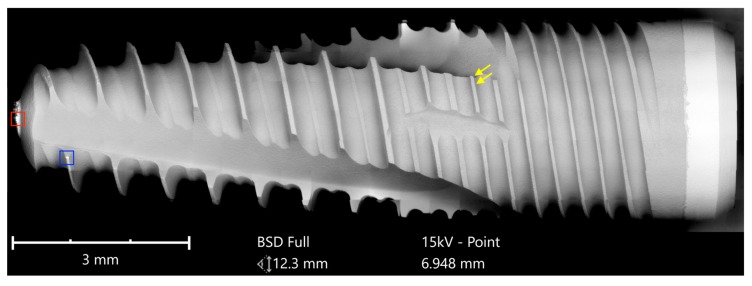
SEM mapping: Dental Implant (Biodenta); yellow arrows indicate spots with aluminum as shown in Figure 21. Magnifications of the blue and red marked areas are shown in Figure 23.

**Figure 20 jcm-08-01280-f020:**
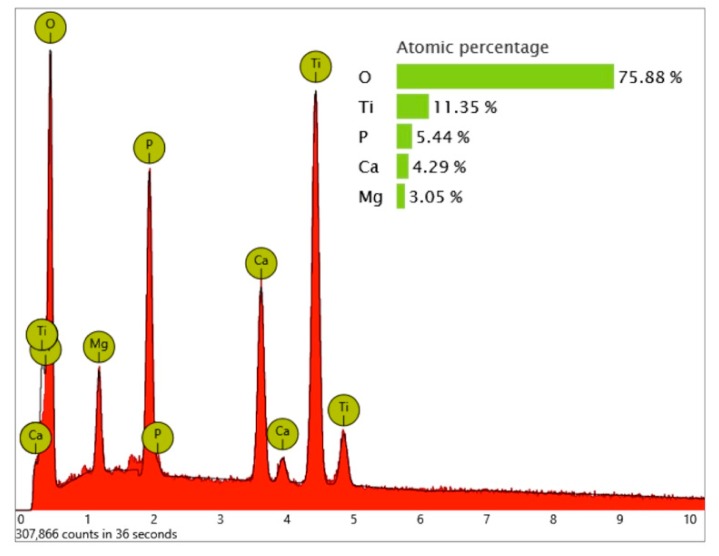
EDS area analysis of the implant’s core material (Biodenta).

**Figure 21 jcm-08-01280-f021:**
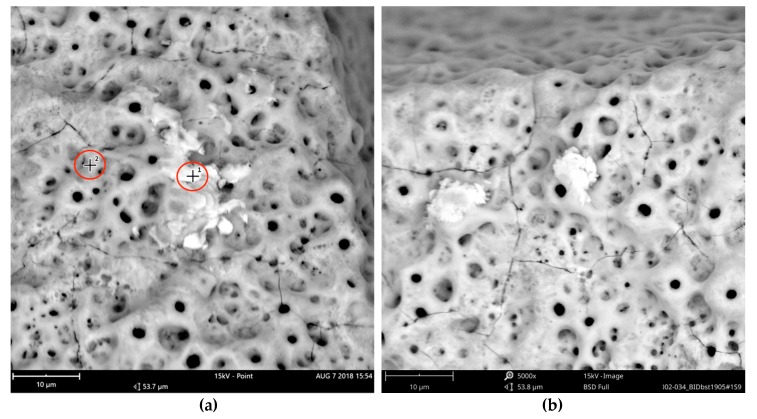
Aluminum-containing particles (5–10 µm) on the surface of the Biodenta implant; location shown using yellow arrows in Figure 19; (**a**) SEM 5000× with 2 points of EDS spot measurement shown in Figure 22; (**b**) impurities with similar size in close proximity to particles shown in the left image (**a**).

**Figure 22 jcm-08-01280-f022:**
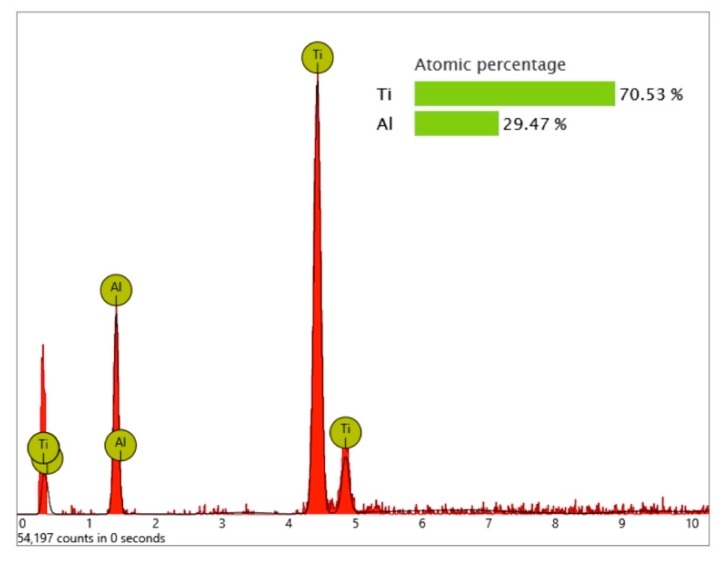
EDS differential measurement of the metal particle in Figure 20 (**a**), showing significant X-ray quanta of aluminum. The lack of oxygen signals indicates that these particles were not made of aluminum oxide.

**Figure 23 jcm-08-01280-f023:**
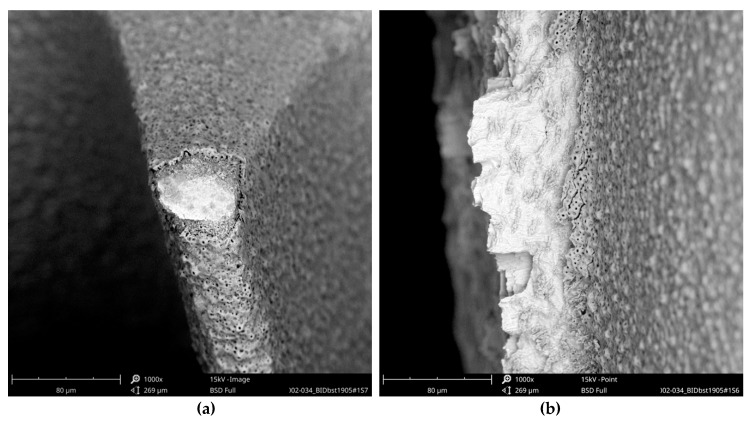
Damaged oxide layer of the Biodenta implant, SEM 1000×: (**a**) magnification of the blue marked area in Figure 19 showing the damage at the exposed implant thread; (**b**) magnification of the red marked area in Figure 19 demonstrates the large-area irregularity of the oxide layer at the implant apex.

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
