# Peer review of "On the Cleanliness of Different Oral Implant Systems: A Pilot Study"

_jcm, 2019, doi:10.3390/jcm8091280_

Round 1

Reviewer 1 Report

This manuscript deals with a matter of concern. Implants are being used without sufficient guarantees. It is necessary to take care of the materials used in Medicine, especially those to be implanted. Adequate levels of quality and cleanliness of materials must be demanded, and users must be informed about what is being implanted. 

Research is well planned, results are clearly shown and conclusions are based on facts. 

I would suggest extending the study not only to the cleanliness of the implants, but also to the quality of the base material.

Author Response

I would suggest extending the study not only to the cleanliness of the implants, but also to the quality of the base material.

Answer: We added ” - A pilot study“ as subtitle to the manuscript. We agree with the reviewer, that there is room for an extension of this research in terms of a more holistic approach, i.e., increasing not only the numbers of samples for each manufacturer - as proposed by reviewer #2 - but also having an additional focus on the core material of the dental implants.

Note: A PDF file in low resolution is attached and shows all corrections (text corrections requested from review #3 are marked in brown color, text corrections on request from review #2 are all marked in blue).

Note: All captations have been revised as requested from reviewer #2. All SEM mapping images (Fig. 3, 5, 11, 13, 17, and 19) were revised and regions of interest were marked, as proposed by reviewer #2. Areas, where the magnification of the subsequent images is 5,000x were marked with an arrow; areas with smaller magnifications (respectively larger field of views) were marked as squares with correspondent sizes.

Page 10 Ln 255-260: (Revision of Fig. 14 and the correspondent captation): In the Discussion (Page Ln 349) we mentioned that we found “two” small particles containing iron on the Bioconcept implant, whereas in the Results section we only mentioned “one metal particle…” (Ln 262) that was shown in Figure 15. In fact, the second particle with iron signals was the organic particle in Figure 14. Thus, by improving all captations, we added this information in the captation of Figure 14 and pointed inside the image at the bright metal material, embedded in the organic particle, with arrows.

THANK YOU FOR YOUR REVIEW AND VALUABLE COMMENTS.

Reviewer 2 Report

In this paper, authors used SEM (BS detector – for images and x-ray detector for composition analyses) as the only tool to compare dental implants in terms chemical composition of the surface. It was used only 6 different implants (the difference among the implants are the manufacture companies) which were divided in three groups, in each group one of implants was identified as the well documented and the other one was identified as the “look-like” implant. Although authors take some conclusions, my first concern is about statistics. It is impossible to make any statement using only one implant.

My advertisement is to suggest authors to analyze at least three implants from each manufactures and to use different technics to analyze the substances found on/in the surface. The chemical composition is not enough to characterize those substances.

Beside this main observation i will present here other small observations:

Ln 93 – authors should add a scale to this image

Ln98 and 99 – instead of magnification, authors should use a resolution parameter to qualify the microscope. Over the manuscript, authors should not mention the magnification but they should use the scales instead (already showed in most of the images).

Ln 115 – for EDS analyses, it was used a single spot as mention on workflow (figure 2). Why authors do not have EDS values for big areas? Usually, EDS analyses are done on a global area and later (zoom in) on a small spot. It is known that the composition ration changes but it helps to find other chemical elements not detected on a small spot analysis.

Ln 123 e 124 – where are the results from roughness?

Ln 160 – Since authors present the results of EDS analysis for the particle they could also present the results for the areas around (background; core) like fig 8.

Ln 160 – the caption used in figure 4 a) and b) are useless. Authors have the scale on both image. However, authors should use the caption to explain what can be observed in the image (p.e. an arrow pointing on particles)

Ln 174 – can you identify the area on the implant surface (fig 5) where fig 6 a) was obtained from – (zoom in area)? Like what is done with fig 6 b) on fig 6 a). Same question for similar images below in the manuscript.

Ln 175 – captions of Figure 6 a) and b) are useless. The same statement for all figures that mention the magnification. In fact, I was expecting that captions could explain those images. Where are the EDS results from images on fig 6? Are they the same as presented in fig 8?

If we look at the scale (fig 6 a) some of organic areas are big so I would like to see those areas identified on fig 5 as well

Ln 203 – how can authors identify those particles as PTFE? The ratio of C:F on PTFE is 1:2 and authors got 2.5:1 (almost the opposite)!

Ln 209 – the meaning of grade 4 should be specified in composition (Ti 99,99999999????).

Ln 213 – fig 11 shows some dark dots on the image. Do you have the results from EDS analysis for those?

Ln 240 – I am very surprised about the purity of this particle identified on fig 16. First because is very unusual to use pure iron (100.00%). Second, pure iron usually forms an oxide under atmosphere at the surface. Did authors detect other small particles like this one on the implant surface?

Whole this study is only based on SEM images and EDS analysis which makes the manuscript very poor. On the other, authors do not describe much about what they observe on the implant surface (all figure captions should be improved). The absence of global surface EDS analyses for all implants compromise discussion and conclusions.

Ln 263 – titanium anodization produces titanium oxide. Were these implants coated with calcium phosphate after? How do authors explain the high amount of Ca, P, O and simultaneously Ti and Mg in the EDS results?

Ln 276 – fig 22 corresponds to the analysis on #1 or #2 on fig 20a)?

Ln 277 – how can these particles be under phagocytosis if they look embedded on the surface?

Author Response

Note: A PDF file in low resolution is attached and shows all corrections regarding this review marked in blue color. (Text corrections in brown were made on request of review #3).

Point 1: My advertisement is to suggest authors to analyze at least three implants from each manufactures and to use different technics to analyze the substances found on/in the surface. The chemical composition is not enough to characterize those substances.

Response 1: Our analysis is not aimed at a statistical comparison between each one of the documented implant systems and each one of the look-alike ones. Instead, we write in the conclusion only that the three documented systems together were clean in comparison to three non-documented systems together. However, we agree with the reviewer that we would have had an improved study if we had gone the extra mile to analyze more implants from each manufacturer.

We have, therefore, changed our head title by adding a subtitle "...-  A pilot study". We have further commented on the lack of statistical analysis in the text by writing at the end of the introduction:

“Every single dental implant has to be clean, as this is a medical device that could harm patients – even if we find one single implant with impurities, this implant was intentionally sold for the therapy of one real patient. This paper was not intended to show a statistically relevant number of average contaminations for specific implant types. All implants in this study were randomly purchased and labeled for clinical use. Each sample of these medical devices is produced using a certain regime of quality management. If the quality management of a manufacturer cannot ensure a certain level of cleanliness, a single implant with significant impurities, which are technically avoidable, is proof of a lack of quality.”

Beside this main observation i will present here other small observations:

Point 2: Ln 93 – authors should add a scale to this image

Response 2: This is a photographic image of the situation in the cleanroom. We added the length of the implant sample in the correspondent caption.

See Correction Page 3 Ln 103:  Figure 1. Implant sample with a length of 10 mm mounted on the SEM sample-holder.

Point 3: Ln98 and 99 – instead of magnification, authors should use a resolution parameter to qualify the microscope. Over the manuscript, authors should not mention the magnification but they should use the scales instead (already showed in most of the images).

Response 3: The resolution of the SEM used for this research project goes down to 15 nm. This SEM allows magnification of up to 100.000. However, the maximum magnification used for this paper was 5000x, which allowed to perform very sharp images and to identify particles and spot analyses of approx. 1 µm. In order to reproduce the findings of scanning electron microscope images, the magnification is relevant. Scales are shown on all SEM images.

See correction Page 3 Ln 108-109: The high-sensitivity BSE-detector allows a magnification of up to 100,000x with a resolution down to 15 nm. This study used material-contrast images from 500x to a magnification of 5,000x.

Point 4: Ln 115 – for EDS analyses, it was used a single spot as mention on workflow (figure 2). Why authors do not have EDS values for big areas? Usually, EDS analyses are done on a global area and later (zoom in) on a small spot. It is known that the composition ration changes but it helps to find other chemical elements not detected on a small spot analysis.

Response 4: We routinely performed complex mapping images of material contrast from backscattered electrons in 500x magnification, showing the complete sample from shoulder to apex in highest resolution. These mapping images of up to 600 single SEM tiles worked as an orientation and map in order to locate spots of different material contrast. Subsequently, we zoomed into these regions of interest and performed small spot imaging and correspondent EDS analysis. As an additional routine procedure, we performed EDS analysis of larger areas for each implant in 2500x in order to verify the signals of the core material for subsequent differential measurements. As the focus of this paper was not to show or compare the different compositions of implant core materials, we did not present these additional results in the paper.

Point 5: Ln 123 e 124 – where are the results from roughness?

Response 5: In this pilot study we only evaluated one implant of each type with respect to surface roughness, which is not how we work conventionally when we have minimally three implants of each type analyzed. However, we had satisfactory data to conclude that all implants were in the moderately rough range. We have changed the writing like this:

See correction Page 4 Line 134-135: All implants seemed to be in the moderately rough surface range, i.e. with Sa values of between 1 and 2 micrometers.

Point 6: Ln 160 – Since authors present the results of EDS analysis for the particle they could also present the results for the areas around (background; core) like fig 8.

Response 6: Figure 8 was shown to explain the principle of the differential measurement in the framework of a spot analysis. All other EDS graphs are focusing on specific pollution. We added a more substantial explanation to the caption of Figure 8:

See Page 7 Line 209-214: EDS differential measurement of organic particle in Figure 7. Graph #1 shows titanium as core material of this implant with characteristic peaks at 0.452 KeV (La), 4.508 KeV (Lb) and 4.932 KeV (Kb). Graph #2 shows the spectrum of the particle with additional signals of the titanium background. The subtraction of X-ray quanta from the background material reveals the elemental composition of the impurity, which is carbon (characteristic Ka peak at 0.277 KeV) and oxygen (Ka peak at 0.523 KeV).

Point 7: Ln 160 – the caption used in figure 4 a) and b) are useless. Authors have the scale on both image. However, authors should use the caption to explain what can be observed in the image (p.e. an arrow pointing on particles)

Response 7: The caption explains that image (b) is a magnification of the white square shown in the center of the image (a). The additional information for image (b) is the spot analysis of a TiOparticle in this image (black cross in the red circle). Instead of arrows, we used red circles for better orientation around the point of EDS analysis, indicated as crosshairs. We added more substantial information to the caption of Figure 4:

Page 5 Line 174 – Figure 4: OsseoSpeed surface; (a) marked area of Fig. 3 in 500x mag.; (b) higher magnification (2,500x) of marked area in the left image with EDS spot-analysis of embedded TiO2-particle (spot is marked with “+” in red circle) showing only signals of the blasting material titanium oxide).

Point 8: Ln 174 – can you identify the area on the implant surface (fig 5) where fig 6 a) was obtained from – (zoom in area)? Like what is done with fig 6 b) on fig 6 a). Same question for similar images below in the manuscript.

Response 8: We thank the reviewer for this suggestion and have followed this suggestion by changing the following figures: (Figs. 3, 5, 11, 13, 17, and 19). Regions of interest were marked, as requested. We used different colors for multiple ROIs. Areas, where the magnification of the subsequent images is 5,000x were marked with an arrow; areas with smaller magnifications (respectively larger field of views) were marked as squares with correspondent sizes.

Point 9: Ln 175 – captions of Figure 6 a) and b) are useless. The same statement for all figures that mention the magnification. In fact, I was expecting that captions could explain those images. Where are the EDS results from images on fig 6? Are they the same as presented in fig 8? YES

Response 9: All captions have been revised as requested. The caption in figure 6 explains the findings and location, as we wrote: “…organic particles (10-50 µm) at the implant shoulder”. We do not understand, why this explanation of

1) the chemical nature (organic),
2) the size (10-50 µm) and
3) the location (shoulder)

was rated in the peer-review as “useless caption”. We understand this information as an appropriate explanation within the limitation of a capitation.

Page 6 Ln 189-192: We revised the caption of Fig. 6a+b: (a) Systematic contamination of exposed threads, red marked area of Fig. 5 in 500x.; (b) Magnification (5,000x) of white marked area in Fig. 6 (a); EDS differential measurement of marked spots is identical with data of particles in Fig. 7.

Point 10: If we look at the scale (fig 6 a) some of organic areas are big so I would like to see those areas identified on fig 5 as well

Response 10: In the high-resolution version of the original image (provided for the printing process) the particles in figure 6 can be seen on the threads 1-7 of the implant. In the correction, the area is now marked, following the proposal of the reviewer and shown in Fig. 5.

Point 11: Ln 203 – how can authors identify those particles as PTFE? The ratio of C:F on PTFE is 1:2 and authors got 2.5:1 (almost the opposite)!

Response 11:  The quantitative information about atomic percentage provided by EDS analysis does not always reflect the precise stoichiometric relationship of the particle´s chemical elements. EDS analysis does not give chemical information, e.g., of the oxidation state or the chemical bonds.

Newbury and Ritchie wrote in an article published 2012 in the Journal of Scanning Microscopies with the title: Is Scanning Electron Microscopy/Energy Dispersive X‐ray Spectrometry (SEM/EDS) Quantitative?: “The use of standardless analysis… leads to quantitative results that, while useful, do not have sufficient accuracy to solve critical problems, e.g. determining the formula of a compound”. (https://doi.org/10.1002/sca.21041) 

Thus, the authors did not “identify” the particles as PTFE. However, we used our word carefully and wrote that texture and elemental composition (i.e, not the elemental stoichiometry) “…suggest that these particles are most likely remnants of polytetrafluoroethylene (PTFE)…”. The careful wording “most likely” is justified, as the material PTFE gets in direct contact with the implant’s surface. PTFE is used in baskets for the acid etching process and sometimes for the coping of the implant’s inner geometry that protects the inner compartment against blasting and acid etching procedures after the turning process.

Page 8 Ln 228 caption Fig. 10 - We added the information: “Note: The quantitative information provided by EDS analysis does not always reflect the precise stoichiometric relationship of the particle´s chemical elements.”

Point 12: Ln 209 – the meaning of grade 4 should be specified in composition (Ti 99,99999999????).

Response 12: We detailed the term as requested:

Page 9 Ln 233-235:“...made of commercially pure grade 4 titanium, which is a composition of 99% titanium, 0.50% iron (max.), oxygen 0.40% (max.), carbon 0.085 (max.), hydrogen 0.015% (max.) and nitrogen 0.05% (max.), according to ASTM F67 specification.”

Point 13: Ln 213 – fig 11 shows some dark dots on the image. Do you have the results from EDS analysis for those?

Response 13: What appears as “dark dots” in Figure 8 can be seen as grooves - not particles - in higher magnification. Figure 11 shows the full-size implant in very high resolution, digitally composed from up to 400 single SEM images in a magnification of 500. The original high-resolution image, which is not comparable to the low-resolution image in the peer-review manuscript, gives proof of this statement. If necessary, we can provide the 16 MB original TIFF image for the peer-review.

Point 14: Ln 240 – I am very surprised about the purity of this particle identified on fig 16. First because is very unusual to use pure iron (100.00%). Second, pure iron usually forms an oxide under atmosphere at the surface. Did authors detect other small particles like this one on the implant surface?

Response 14: X-rays are emitted throughout the volume of material into which the electron beam is scattered and decelerated, so the resolution is about 1-2 micrometers in depth. The thickness of the iron oxide layer is in the nanometer range. Thus, the EDS technique is not suitable to measure substantial amounts of oxygen from any thin oxide layer. XPS would be the right choice to measure the thickness of any oxide. However, this was not in the scope of this research.

Other traces of iron were found on the same implant as described in the new caption of Fig. 14. However, no other implant (of 250 samples from more than 140 manufacturers) that we analyzed in the past ten years, showed particles like the one in Fig. 15.

Point 15: Whole this study is only based on SEM images and EDS analysis which makes the manuscript very poor. On the other, authors do not describe much about what they observe on the implant surface (all figure captions should be improved). The absence of global surface EDS analyses for all implants compromise discussion and conclusions.

Response 15: We disagree with all due respect, that research based only on SEM/EDS data is per se a poor one. SEM imaging and EDS analyses performed sufficient data within the scope of this manuscript. A global surface EDS analysis, which means an analysis of the complete implant from shoulder to the apex is technically not possible with accurate data, because a long distance is necessary to see the complete sample in the SEM. This long distance, however, does not allow collecting a sufficient amount of x-ray quanta for the EDS detector.

Point 16: Ln 263 – titanium anodization produces titanium oxide. Were these implants coated with calcium phosphate after? How do authors explain the high amount of Ca, P, O and simultaneously Ti and Mg in the EDS results?

Response 16: The FDA premarket notification 510(k) submission of the Biodenta Dental Implant System (dated March 19, 2013) only mentioned a “spark anodization” for the surface treatment. No source was found in the literature, in the product information or on the company´s website, that an additional coating with calcium phosphate or magnesium was performed.

Point 17: Ln 276 – fig 22 corresponds to the analysis on #1 or #2 on fig 20a)?

Response 17: Neither nor. As described in the new caption of Fig. 22 the graph shows a differential spectrum, i.e. the difference of the spectrum from spot #1 (= signals from particle + core material) and spot #2 (core material), thus giving information of the particle´s elemental composition without the core material.

Point 18: Ln 277 – how can these particles be under phagocytosis if they look embedded on the surface?

Response 18: We have not written, that they are phagocytized. However, foreign materials that have no chemical bond to the implant’s surface are exposed to high forces of friction during the process of insertion (20-70 Ncm). It is highly likely that part of the material (organic and inorganic particles) is losing mechanical contact with the implant’s surface and will be exposed to the surrounding tissue.  We corrected the text and moved this topic - on request of another reviewer - to the Discussion:

See Page 15 Ln 350-352: These particles, with a diameter of 5 to 10 µm, and organic contaminants with similar size are small enough for phagocytosis by macrophages that would be theoretically possible for particles without a chemical bonding to the implant surface.

Additional Note: Page 10 Ln 255-260: (Revision of Fig. 14 and the correspondent caption)

In the Discussion (Page Ln 349) we mentioned that we found “two” small particles containing iron on the Bioconcept implant, whereas in the Results section we only mentioned “one metal particle…” (Ln 262) that was shown in Figure 15. In fact, the second particle with iron signals was the organic particle in Figure 14. Thus, by improving all captions, we added this information in the caption of Figure 14 and pointed inside the image at the bright metal material, embedded in the organic particle, with arrows.

THANK YOU FOR YOUR REVIEW AND VALUABLE COMMENTS.

Reviewer 3 Report

General comments:

       The study, conducted by experts in the field, reports on contaminants and supporting literature of leading implant systems (Dentsply Implants, Straumann and Nobel Biocare), compared to the look-alike counterparts (Cumdente, Bioconcept and Biodenta). The merit of this study is to provide data on look-alike implants that, despite being clinically used, lack supporting literature. While samples of the established implant systems showed surfaces free of foreign materials, high magnification detected impurities on the surface of look-alike samples. Literature mentioning look-alike implants mainly investigates prosthetic components and lacks data on long-term biological survival and success.

         My only recommendation is to avoid comments in paragraphs other than the Discussion.  

Specific comments:

·         Page 2 line 85-86: in the paragraph Experimental Section, implant prices should be reported without comments, and the sentence “thus, indicating that the definition of a look-alike or copy-cat product is not necessarily based on a low price.” should be delated or mentioned into the Discussion.

·         Page 14 line 277-278: the sentence “These aluminum particles are small enough for phagocytosis by macrophages with all subsequent consequences” should be moved from the Results into the Discussion, since biological effects of aluminum particles on macrophages is not a result of the study.

·         Page 17 line 367: the comment “all particles small enough for phagocytosis” should be delated from the Conclusions and expanded into the Discussion.

·         Conclusions line 362-372: Please consider re-writing the Conclusions with sentences free from interpretations and based on the Results.

Author Response

Note: A PDF file with the revised version in low resolution is attached and shows all corrections regarding this review marked in brown color. (Text corrections in blue were made on request of Review #2).

Specific comments:

Point 1: Page 2 line 85-86: in the paragraph Experimental Section, implant prices should be reported without comments, and the sentence “thus, indicating that the definition of a look-alike or copy-cat product is not necessarily based on a low price.” should be delated or mentioned into the Discussion.

Response 1: We moved this issue into the Discussion (Page 16 Ln 354-355).
See remaining information on Page 3 Ln 94-96.

Point 2: Page 14 line 277-278: the sentence “These aluminum particles are small enough for phagocytosis by macrophages with all subsequent consequences” should be moved from the Results into the Discussion, since biological effects of aluminum particles on macrophages is not a result of the study.

Response 2: We moved this issue into the Discussion and changed the focus instead only on aluminum particles onto small particles in general, as seen on Page 15 Ln 350-352

Point 3: Page 17 line 367: the comment “all particles small enough for phagocytosis” should be delated from the Conclusions and expanded into the Discussion.

Response 3: We explained this issue more in detail in the Discussion (Page 15 Ln 350-352). However, the authors wanted to focus in the Conclusion also on the size of the particles found in our analyses, i.e., we wanted to emphasize that these particles are a potential risk because - and not despite - they are small, in order to avoid a widespread misconception that small particles don´t matter. Thus, we would recommend keeping the modified phrase " - all particles small enough for possible phagocytosis - " in the  Conclusion (Page 17 Ln 410).

Point 4: Conclusions line 362-372: Please consider re-writing the Conclusions with sentences free from interpretations and based on the Results.

Response 4: We deleted interpretations from the conclusion (Page 17 Ln 406-412).

Note: All captions have been revised as requested from reviewer #2.
In addition, all SEM mapping images (Fig. 3, 5, 11, 13, 17, and 19) were revised and regions of interest were marked, as proposed by reviewer #2. Areas, where the magnification of the subsequent images is 5,000x were marked with an arrow; areas with smaller magnifications (respectively larger field of views) were marked as squares with correspondent sizes.

Additional Note: Page 10 Ln 255-260: (Revision of Fig. 14 and the correspondent caption)
In the Discussion (Page Ln 349) we mentioned that we found “two” small particles containing iron on the Bioconcept implant, whereas in the Results section we only mentioned “one metal particle…” (Ln 262) that was shown in Figure 15. In fact, the second particle with iron signals was the organic particle in Figure 14 (Ln 255). Thus, by improving all captions, we added this information in the caption of Figure 14 and pointed inside the image at the bright metal material, embedded in the organic particle, with arrows.

THANK YOU FOR YOUR REVIEW AND VALUABLE COMMENTS.
